# Fetal Lung Volume Appears to Predict Respiratory Morbidity in Congenital Diaphragmatic Hernia

**DOI:** 10.3390/jcm12041508

**Published:** 2023-02-14

**Authors:** Valentine Cerbelle, Kévin Le Duc, Stephanie Lejeune, Sébastien Mur, Héloise Lerisson, Elodie Drumez, Rony Sfeir, Julien Bigot, Pauline Verpillat, Riadh Boukhris, Pascal Vaast, Clémence Mordacq, Caroline Thumerelle, Laurent Storme, Antoine Deschildre

**Affiliations:** 1Paediatric Pulmonology and Allergy Unit, Hôpital Jeanne de Flandre, CHU Lille, Université de Lille, F-59000 Lille, France; 2ULR2694 Metrics-Perinatal Environment and Health, Université de Lille, F-59000 Lille, France; 3Department of Neonatology, Hôpital Jeanne de Flandre, CHU Lille, Université de Lille, F-59000 Lille, France; 4Pediatric Imaging Unit, Hôpital Jeanne de Flandre, CHU Lille, Université de Lille, F-59000 Lille, France; 5ULR 2694-METRICS: Évaluation des Technologies de Santé et des Pratiques Médicales, CHU Lille, Université de Lille, F-59000 Lille, France; 6Pediatric Surgery Unit, Hôpital Jeanne de Flandre, CHU Lille, Université de Lille, F-59000 Lille, France; 7Jacquemars Giélée Imaging Center, F-59000 Lille, France; 8Générale de Santé, La Louvière Ramsay Hôpital, F-59000 Lille, France; 9Obstetrics and Gynecology Unit, Hôpital Jeanne de Flandre, CHU Lille, Université de Lille, F-59000 Lille, France

**Keywords:** congenital diaphragmatic hernia, fetal magnetic resonance imaging, fetal lung volume, antenatal diagnosis, prognostic factor, respiratory morbidity, child

## Abstract

Congenital diaphragmatic hernia (CDH) is associated with pulmonary hypoplasia and respiratory morbidity. To assess whether respiratory morbidity during the first 2 years of life in infants with left-sided CDH is associated with fetal lung volume (FLV) evaluated by the observed-to-expected FLV ratio (o/e FLV) on antenatal magnetic resonance imaging (MRI). In this retrospective study, o/e FLV measures were collected. Respiratory morbidity in the first 2 years of life was studied according to two endpoints: treatment with inhaled corticosteroids for >3 consecutive months and hospitalization for any acute respiratory disease. The primary outcome was a favorable progression defined by the absence of either endpoint. Forty-seven patients were included. The median o/e FLV was 39% (interquartile range, 33–49). Sixteen (34%) infants were treated with inhaled corticosteroids and 13 (28%) were hospitalized. The most efficient threshold for a favorable outcome was an o/e FLV ≥ 44% with a sensitivity of 57%, specificity of 79%, negative predictive value of 56%, and positive predictive value of 80%. An o/e FLV ≥ 44% was associated with a favorable outcome in 80% of cases. These data suggest that lung volume measurement on fetal MRI may help to identify children at lower respiratory risk and improve information during pregnancy, patient characterization, decisions about treatment strategy and research, and personalized follow-up.

## 1. Introduction

Congenital diaphragmatic hernia (CDH) is a rare birth malformation involving a diaphragmatic defect that allows the intraabdominal organs to migrate upward into the thorax and which compromises normal lung development. In 2017, the prevalence rates per 10,000 live births in Europe were 2.47, and this rate increased to 3.45 when including termination of pregnancy for fetal anomalies or death [1]. Lung development depends on many biological as well as mechanical factors [2]. The compression related to the presence of abdominal organs in the intrathoracic space, particularly the liver in right-sided CDH, can cause parenchymal and pulmonary vascular hypoplasia. The position of the liver is a predictor of survival [3,4].

The respiratory consequences of CDH in children are heterogeneous and can include very early death, need for oxygen therapy on day 28, which is defined as chronic lung disease (CLD), persistent need for oxygen therapy, no neonatal complications, and even delayed diagnosis. Neonatal care is also heterogeneous and can be affected by the caseload, the team’s experience, and neonatal facilities; this had led to the centralization of the treatment of patients with this rare disease in some countries.

Identifying prognostic factors in the prenatal period remains an important issue. The use of magnetic resonance imaging (MRI), as proposed by the French “Protocole National de Diagnostic et de Soins” has led to significant progress in the evaluation of fetal pulmonary hypoplasia [5]. A fetal MRI is now currently performed when CDH is suspected on ultrasound. MRI allows the direct measurement of fetal lung volume (FLV) and the calculation of the observed-to-expected FLV ratio (o/e FLV) for gestational age [6]. In a meta-analysis, Oluyomi-Obi et al. [7] examined the performance of o/e FLV for predicting neonatal mortality in CDH. This ratio has been shown to predict the need for extracorporeal membrane oxygenation (ECMO) or CLD [8,9,10]. Increasing knowledge about the antenatal prognosis, especially long-term respiratory morbidity, could help improve the information given to parents during pregnancy, predict the risk of infants with CDH, and inform the planning of therapeutic options, follow-up care, and research.

To our knowledge, no studies have evaluated the prenatal prognostic factors for respiratory outcomes during the first years of life. The main objective of our study was to identify a threshold value of o/e FLV that is associated with a better prognosis for respiratory morbidity during the first 2 years of life in a cohort of children with left-sided CDH born between 2009 and 2015 at the French Reference Centre for CDH, Lille University Hospital. The secondary objective was to examine the relationships between the o/e FLV measured on MRI and acute or chronic respiratory morbidity.

## 2. Material and Methods

Our study was carried out at Lille University Hospital in the French reference center for CDH.

All patients born between 1 January 2009 and 31 December 2015 with a prenatal diagnosis of left-sided CDH, with at least one fetal MRI performed, and who remained alive at 2 years of age were included. We conducted a retrospective analysis of the data collected prospectively during the follow-up of these children between birth and 2 years of age using a standardized questionnaire. The data were an integral part of the issues routinely addressed during follow-up visits by neonatologists and recorded in medical reports; they were collected from September 2017 to March 2018. The study was approved by the French Data Protection Authority (CNIL) (No. DEC19-328).

### 2.1. Data Collection

Data on sex, gestational age expressed in weeks, premature birth, and type of CDH (intrathoracic liver, patch repair) were collected from the local database. The type of ventilation support (mechanical ventilation, high-frequency oscillatory (HFO), or noninvasive ventilation), the need for ECMO, the total duration (days) and need for supplemental oxygen at day 28 (defined as CLD), were recorded for each patient.

### 2.2. Patient Care

Management of the CDH infants is based on the EURO Consortium guidelines and the French Reference Center guidelines [11,12]. Newborn infants are mechanically ventilated in pressure-controlled mode (STEPHANIE; Fritz Stephan GmbH Medizintechnik, Gackenbach, Germany) with an inspiratory pressure between 18 and 24 cmH_2_O and a positive end-expiratory pressure (PEEP) between 2 and 5 cmH_2_O. The fraction of inspired oxygen (FiO_2_) and ventilator parameters are adjusted to target pre-ductal SpO_2_ 85–95% and PaCO_2_ 45–60 mmHg. The use of high-frequency oscillatory ventilation (HFOV) can be considered a means of reducing baro-volotrauma. It is indicated in situations of persistent hypercapnia (PaCO_2_ > 60 mmHg) and/or hypoxemia (preductal SpO_2_ < 85%, below 1 FiO_2_). ECMO is used in CDH infants in the following situations: preductal SpO_2_ < 85% while inspiratory pressure is > 28 cmH_2_O (or mean pressure > 16 cmH_2_O in HFOV) and signs of circulatory failure; or oxygenation index > 40 for more than 6 h; or occurrence of refractory intercurrent septic shock.

### 2.3. Diagnosis of Respiratory Morbidity

To study the respiratory outcomes during the first 2 years of life, respiratory morbidity was defined according to two criteria: (1) daily treatment with inhaled corticosteroids (ICS) for >3 consecutive months and (2) occurrence of at least one unscheduled hospitalization for lower respiratory tract illness. These two objective criteria allowed a global approach to quantifying respiratory morbidity by using data about the daily treatment, which reflects chronic respiratory morbidity, and hospital admission, which reflects severe acute morbidity. These two criteria have been studied separately in the literature. In 2001, Muratore et al. [13] reported that 35% of 100 children with a history of CDH required at least transient therapy during the first year. Benoist et al. [14] studied hospitalizations for respiratory reasons and found that 22 of 86 children (26%) were readmitted at least once for wheezing exacerbations before 2 years of age. We defined a favorable progression as the absence of these two criteria and unfavorable progression as the presence of one or both criteria.

### 2.4. Fetal Lung Volume Measurement on MRI

As part of the follow-up performed at the CDH reference center, one or two fetal MRI imaging scans were performed after the prenatal diagnosis of CDH: the first between 26 and 28 weeks and the second between 30 and 32 weeks of gestation. In cases of two MRIs, we used the first images. The fetal MRIs were performed using a Philips Ingenia 3 TESLA MRI imaging scanner. No premedication was given, and the mother was placed in supine position. The examination time was about 30 min. The protocol included three T2 planes (TE 70) and T1-weighted sequences in the axial and sagittal planes. The T2-weighted images were obtained using a single-shot half-Fourier TSE sequence in the orthogonal transverse coronal and sagittal planes according to the fetal orientation. The sequences affected by fetal movements were repeated. The cutting thickness was 4 mm. Measurements were performed by two radiologists qualified for this evaluation using PACS software (IntelliSpace PACS 4.4, Philips, Amsterdam, The Netherlands) using a contour (freehand region of interest) cut. The contouring was performed on the sequence that allowed the best visualization, either coronal or sagittal (Figure 1). The o/e FLV was calculated as a function of gestational age using the Rypens formula and is expressed as a percentage [6].

### 2.5. Statistical Analysis

Qualitative variables are reported as a number (percentage) and continuous variables as a median (interquartile range (IQR)). The normality of the distributions was assessed graphically and using the Shapiro–Wilk test. The o/e FLV was compared between two groups of patients using Mann–Whitney’s *U* test. The optimal threshold value of the o/e FLV for predicting a favorable respiratory outcome was determined by maximizing the Youden index. Sensitivity, specificity, and positive and negative predicted values for the optimal threshold value were calculated. All statistical tests were performed at the two-tailed *α* level of 0.05 using SAS software (release 9.4; SAS Institute, Cary, NC, USA).

## 3. Results

### 3.1. Study Population

From 2009 to 2015, 62 prenatal diagnoses of left-sided CDH were made. Fifteen patients (24%) were excluded, including two who had no fetal MRI, three who were lost during the systematic follow-up, and ten who died before two years of age. The characteristics of these 10 children and the causes and times of their deaths are listed in Table 1. Finally, 47 patients (76%) who were still followed at 2 years of age were included (Table 2). The median o/e FLV was significantly lower in the deceased than in the living patients (29% (IQR, 26–38) versus 39% (IQR, 33–49); *p* = 0.047). The median durations of mechanical ventilation and oxygen therapy (all interfaces combined) were 5 days (IQR, 3–9) and 14 days (IQR, 5–34), respectively. Nine (19%) patients required HFO and two (4%) ECMO. A total of 13 (28%) patients were diagnosed as having CLD. During the hospitalization period, six patients (13%) were diagnosed as having ventilator-associated pneumonia. Four (8%) had a pleural effusion and four had a pneumothorax. Only one patient had chylothorax during his hospitalization.

### 3.2. Respiratory Morbidity

Sixteen patients (34%) were treated for >3 months with ICS. Thirteen (28%) patients were hospitalized at least once for an acute respiratory tract illness during the first 2 years of life: eleven (23%) in the first year, four (9%) in the second year, and two (4%) at least once a year. The median o/e FLV according to the two criteria are shown in Table 3. The o/e FLV was lower in patients treated with ICS than in those who were not (33% (IQR, 28–41) versus 44% (IQR, 34–58); *p* = 0.027). The o/e FLV did not differ between patients who were hospitalized at least once for respiratory reasons and those who were not (34% (IQR, 32–41) versus 42% (IQR, 33–52); *p* = 0.082).

### 3.3. Optimal Threshold Value of o/e FLV and Respiratory Morbidity

In total, 28 patients (60%) had a favorable outcome, and 19 (40%) had an unfavorable progression (Table 3). The median o/e FLV was higher in patients with a favorable outcome (45% (IQR, 35–59)) than in those with an unfavorable progression (34% (IQR, 29–42); *p* = 0.024).

The optimal threshold value of o/e FLV to predict a favorable respiratory outcome was 44% with a sensitivity of 57%, specificity of 79%, negative predictive value of 56%, and positive predictive value of 80%. In other words, an o/e FLV ≥ 44% was associated with a favorable outcome in 80% of cases.

## 4. Discussion

Our study reports on the relationship between lung volume measured by MRI and respiratory morbidity during the first 2 years of life in infants with left-sided CDH. We found that o/e FLV may be useful for identifying patients with a lower risk of respiratory morbidity.

Patients with CDH are at risk of pulmonary impairment, although the respiratory risk is heterogeneous. Therefore, it is crucial to identify factors that can help predict the respiratory outcome of these children. Several studies have searched for associations with the mortality or morbidity of different combinations of prognostic factors by focusing on the possible associations between antenatal markers, for example, the associations between o/e FLV and the development of CLD or between postnatal markers such as the dosage of N-terminal pro-brain natriuretic peptide and the need for ECMO support [7,9,15].

We chose to study o/e FLV, an antenatal marker, using MRI because this technique has many advantages over ultrasound. MRI allows pulmonary visualization in all three planes and a complete view of all thoracic structures, particularly the position of the liver. The study by Büsing et al. [16] demonstrated the validity and reliability of lung volume measurements by MRI independent of the selected analytical sequences in 40 fetuses with CDH. Other studies have shown the superiority of MRI as a predictor of survival compared with ultrasound [3,17,18]. Abbasi et al. [19] evaluated the measurement of the lung-to-head ratio (LHR) in 26 centers in the US and found variability in the ultrasound method; they concluded that this measurement needs to be standardized. In a recent study of 42 patients with left-sided CDH, Kim et al. [18] found that o/e FLV was more accurate than prenatal ultrasound for predicting the severity of pulmonary hypoplasia and discerning survivors from nonsurvivors. With the continued development of MRI, some researchers have sought to establish a validated prognosis using new factors. For example, Cannie et al. [20] evaluated the relationship between postnatal survival and the liver-to-thoracic volume ratio measured with MRI and found a correlation in 30 fetuses without tracheal occlusion.

Many studies have pointed out that o/e FLV is a prognostic factor for mortality [7,16,21,22,23,24]. Cordier et al. reported a link between o/e FLV and mortality: an o/e FLV < 15% was associated with mortality in 88% of cases with upward liver migration. In a meta-analysis that included 22 studies published up to 2016, Oluyomi-Obi et al. [7] examined the performance of o/e FLV for predicting neonatal mortality in CDH. They reported that with the same o/e FLV threshold of 25%, the survival rates were 0–25%. This meta-analysis also found that eight studies reported a statistically significant difference between the o/e FLV in survivors compared with deceased patients. We found a similar result: an o/e FLV of 39% in survivors versus 29% in the others (*p* = 0.047). In another study by Cordier et al., o/e FLV > 45% was associated with a survival rate better than 75% [24]. To our knowledge, no previous work has investigated the correlation between prenatal prognostic factors and respiratory outcomes after one year.

Recent studies have focused on FLV as a predictor of respiratory morbidity beyond the neonatal period, including the development of CLD or the need for oxygen therapy at 1 year of age [9,25]. In a large study of 148 patients with left-sided and 22 with right-sided CDH, Debus et al. [9] found that fetuses with an o/e FLV of 50% had only a 4.5% risk of developing CLD of any grade and a 0.2% risk of developing severe CLD. Tsuda et al. [25] calculated a threshold value for predicting the need for oxygen therapy at 1 year of age in 41 patients with CDH, including 3 with right-sided CDH. The o/e FLV threshold was 25% of PV, with a sensitivity of 88.6% and a specificity of 40.0%.

To our knowledge, no study has examined the link between o/e FLV and long-term respiratory outcomes. It is crucial to be able to identify patients at risk for adverse outcomes as early as the antenatal period, as well as those most likely to have a positive outcome, by determining a threshold associated with lower chronic and acute respiratory morbidity; this was our aim in the present study. Identifying a threshold is important for improving information given to parents and for making decisions about treatment and outpatient follow-up care. In our study, an o/e FLV ≥ 44% of PV was associated with a favorable outcome in 80% of cases. By contrast, Debus et al. [9] reported that this ratio was associated with a risk of antenatally estimated probability of CLD of any grade at day 28 of 9.7%, and mainly mild CLD.

We found a significant relationship between o/e FLV and chronic respiratory morbidity, as reflected in the long-term prescription of ICS. Chronic morbidity and ICS prescription are frequently observed in patients with CDH [13,26]. Muratore et al. [13] found that 40% of a population of 100 patients born with CDH required ICS treatment during the first year of life, although their study did not report the duration of treatment. Cauley et al. [27] studied whether a diagnosis of asthma at 5 years of age was related to the need for pulmonary support at day 30 in 105 patients with CDH. Patients on pulmonary support had a significantly greater use of steroids and a likelihood of asthma at 5 years of age.

We found that the association link between o/e FLV and severe acute respiratory exacerbations leading to hospitalization was not significant. Respiratory complications leading to hospitalization occur most frequently in the first 3 years of life [14,28]. Benoist et al. [14] collected data prospectively on readmissions before 2 years of age. In 86 children with CDH born between 2009 and 2013, 22 (26%) were readmitted at least once for respiratory exacerbation; our results are consistent with this rate of readmission. The small size of our population may explain the lack of a significant association between o/e FLV and readmission.

### Strengths and Weaknesses of the Study

This study was conducted in only one reference center that provided homogeneous and expert management of patients with CDH. This study also included children born between 2009 and 2015, i.e., over a period of 6 years marked by continuity in the management strategy and without major changes. We included only children with a left-sided hernia because right-sided CDH occurs less frequently but has a more severe course and a higher risk of respiratory morbidity [29,30,31]. However, a recent French study found that right-sided CDH does not have a higher risk of mortality than left-sided CDH after adjusting for the LHR and upward migration of the liver [31]. A comparison of respiratory morbidity between left- and right-sided CDH during the first 2 years of life would be interesting, although difficult to perform given their difference in prevalence.

Our study has some limitations. The number of patients was small because of the low prevalence of this rare disease, and this contributed to the low power of the statistical analysis. The small number of patients in this study is probably responsible for the lack of sensitivity of the o/e FLV to predict the risk of respiratory morbidity later in life. The performance of our threshold was moderate, especially in terms of sensitivity and negative predictive value. It will be crucial to confirm the association between o/e FLV and respiratory morbidity in a larger population. Nevertheless, our results can be helpful to the clinician to orient the parental antenatal information. A larger study would allow the prognostic value of this marker to be determined more accurately. The association between CDH and morbidity should be explored over a longer term, including childhood and adolescence. As CDH is associated with impaired lung development, an assessment of the association between lung volume and long-term respiratory morbidity, including lung function, is needed to identify high-risk populations who may benefit from multidisciplinary and personalized management.

## 5. Conclusions

CDH is a complex disease that has significant heterogeneity in its morbidity. Our findings support the use of FLV as an easily measured component during pregnancy to predict respiratory morbidity during the first year of life. We suggest using a threshold value to identify infants likely to progress along a more favorable course. These data must be confirmed and validated in a larger population. Longitudinal follow-up studies are needed to assess the morbidity during childhood and adolescence. These data are important for improving parents’ access to information during pregnancy and for distinguishing low- from high-risk patients.

## Figures and Tables

**Figure 1 jcm-12-01508-f001:**
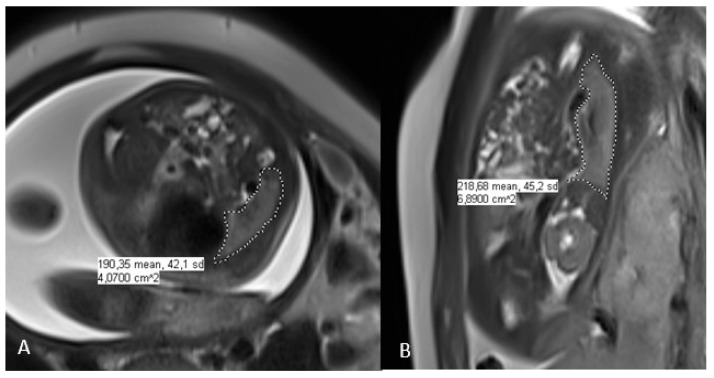
MRI assessment of the right CDH allows the calculation of fetal lung volume (FLV); (**A**): axial and (**B**): coronal.

**Table 1 jcm-12-01508-t001:** Characteristics of the 10 excluded children who died before 2 years of age.

Features and Characteristics	Value
Observed/expected FLV (%)	29 [26–38]
Median GW	36 [31–40]
Prematurity	2 (20)
Male	3 (30)
Time of death:	
Before 24 h of life	1 (10)
After 24 h of life, without hospital discharge	6 (60)
Between hospital discharge and 2 years of age	3 (30)

Values are presented as median [interquartile range] or frequency (percentage). Prematurity was defined as a delivery term < 37 GW; FLV: fetal lung volume; GW: gestational weeks.

**Table 2 jcm-12-01508-t002:** Characteristics of the 47 included patients.

Features and Characteristics	Value
Observed/expected FLV (%)	39 [33–49]
Median GW	39 [38 + 5–40]
Prematurity	4 (9)
Male	28 (60)
Intrathoracic liver herniation	16 (34)
Patch repair	15 (32)
Median total O_2_ duration (days)	14 [5–34]
Median mechanical ventilation duration (days)	5 [3–9]
HFO	9 (19)
Median HFO duration (days)	4 [3–7]
ECMO	2 (4)
CLD	13 (28)

Values are presented as median [interquartile range] or frequency (percentage). Prematurity was defined as a delivery term < 37 GW; FLV: fetal lung volume; GW: gestational weeks HFO: high-frequency oscillatory; ECMO: extracorporeal membrane oxygenation; CLD: chronic lung disease.

**Table 3 jcm-12-01508-t003:** Value of o/e FLV as a function of daily ICS treatment, hospitalization, and outcomes.

Features and Characteristics	Value	o/e FLV	*p*
ICS daily treatment < consecutive months			
Yes	16 (34)	33% [28–41]	0.027
No	31 (66)	44% [34–58]	
≥1 hospitalization for acute respiratory illness			
Yes	13 (28)	34% [32–41]	0.082
No	34 (72)	42% [33–52]	
Outcome			
Favorable	28 (60)	45% [35–59]	0.024
Unfavorable	19 (40)	34% [29–42]	

Values are presented as median [interquartile range] or frequency (percentage); FLV: fetal lung volume; ICS: inhaled corticosteroids.

## Data Availability

The data presented in this study are available on request from the corresponding author.

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
