# Peer review of "Fetal Lung Volume Appears to Predict Respiratory Morbidity in Congenital Diaphragmatic Hernia"

_jcm, 2023, doi:10.3390/jcm12041508_

Round 1

Reviewer 1 Report

We are thankful for the opportunity to review this manuscript.

I believe that the authors have clearly shown the impact of FLV on long-term respiratory CDH outcomes.

I would suggest only to stress in the limit the small numeber of patients included which prevent the develpment of a sensitive prognostic cut-off

Author Response

Thank you for your comment, we have added this information to “Strengths and weaknesses of the study” part lines 280 to 286.

“The small number of patients in this study is probably responsible for the lack of sensitivity of the o/e FLV to predict the risk of respiratory morbidity later in life. Nevertheless, our results can be helpful to the clinician to orient the parental antenatal information. A larger study would allow the prognostic value of this marker to be determined more accurately.”

Reviewer 2 Report

To assess whether respiratory morbidity, during the first 2 years of infants with left sided congenital diaphragmatic hernia (CDH) is associated with fetal lung volume (FLV), Cerbelle et al. evaluated by the observed-to-expected FLV ratio (o/e FLV) on antenatal magnetic resonance imaging (MRI) conducted in only one center. They o/e FLV >44% is the most efficient threshold for a favorable outcome , with sensitivity, specificity, and negative predictive value were 57%, 79%, and 56% respectively. An o/e FLV>44% was associated with a favorable outcome in 80% in cases.

Therefore, they concluded that lung volume measurement on fetal MRI may help to identify children at lower respiratory risk and improve information during pregnancy, patient characterization, decisions about treatment strategy and research, and personalized follow up. It is interesting to examine the prognosis of congenital diaphragmatic hernia in the fetal period, there are limitations to the results derived from this study.

Major

1.      According to past reports, there is no doubt that o/e FLV is a prognostic factor for motality, Cordier AG et al. previously reported that an o/e FLV<15% was associated with mortality in 88% of cases with upward migration of the liver (Cordier AG, Seminars in Perinatology, 2019). In this study, an o/e FLV>44% is considered to have good prognostic, but please describe the differences from past reports.

2.      The threshold of FLV > 44%, which was derived to indicate to have a good prognosis in this study, with sensitivity 57% and specificity 79%. is not so high.

3.      Therefore, The title of this study, “Fetal lung volume predict respiratory mobidity in congenital diaphragmatic hernia'' seems to have low sensitivity and specificity.

4.      It is better to post an MRI picture so that it is easy to imagine that the fetal lung volume was measured using MRI.

Author Response

  • According to past reports, there is no doubt that o/e FLV is a prognostic factor for mortality, Cordier AG et al. previously reported that an o/e FLV<15% was associated with mortality in 88% of cases with upward migration of the liver (Cordier AG, Seminars in Perinatology, 2019). In this study, an o/e FLV>44% is considered to have good prognostic, but please describe the differences from past reports.

We thank you for your interesting and pertinent comment.

To answer this comment, we had the following sentences line 228 to 232.

“In another study by Cordier et al, o/e FLV> 45% was associated with a survival rate better than 75% [24]. To our knowledge, no previous work has investigated the correlation between prenatal prognostic factors and respiratory outcomes after one year.”

  • The threshold of FLV > 44%, which was derived to indicate to have a good prognosis in this study, with sensitivity 57% and specificity 79%. is not so high.

Thank you for this comment. Indeed the sensitivity of this marker is not very high.  But it can help the clinician in his information to the parents during pregnancy.

We have nuanced the interest of this marker in the discussion and added your remark in the limits of the study line 280 to 286.

            “The small number of patients in this study is probably responsible for the lack of sensitivity of the o/e FLV to predict the risk of respiratory morbidity later in life. Nevertheless, our results can be helpful to the clinician to orient the parental antenatal information. A larger study would allow the prognostic value of this marker to be determined more accurately.”

  • Therefore, The title of this study, “Fetal lung volume predict respiratory morbidity in congenital diaphragmatic hernia'' seems to have low sensitivity and specificity.

Thank you for this helpful remark, we have changed the title of our study to take this into account.  Our title is now:

“Fetal pulmonary volume appears to predict respiratory morbidity in congenital diaphragmatic hernia”.

  • It is better to post an MRI picture so that it is easy to imagine that the fetal lung volume was measured using MRI.

Thank you for your advice. A Figure have been added in “Fetal Lung Volume Measurement on MRI” part.

The cutting thickness was 4 mm. Measurements were performed by two radiologists qualified for this evaluation using PACS software (IntelliSpace PACS 4.4, Philips) using a contour (freehand region of interest) cut. The contouring was performed on the sequence that allowed the best visualization, either coronal or sagittal [Figure 1]

Round 2

Reviewer 2 Report

The manuscript is well revised.